# TLE3 loss confers AR inhibitor resistance by facilitating GR-mediated human prostate cancer cell growth

Sander AL Palit[1]*, Daniel Vis[1,2], Suzan Stelloo[3], Cor Lieftink[1], Stefan Prekovic[3], Elise Bekers[4], Ingrid Hofland[5], Tonći Šustić[1,2], Liesanne Wolters[1], Roderick Beijersbergen[1], Andries M Bergman[6], Balázs Győrffy[7,8,9], Lodewyk FA Wessels[1,2], Wilbert Zwart[3,10], Michiel S van der Heijden[1,6]*

[1]Division of Molecular Carcinogenesis, Netherlands Cancer Institute, Amsterdam, Netherlands; [2]Division of Molecular Carcinogenesis, Oncode Institute, Netherlands Cancer Institute, Amsterdam, Netherlands; [3]Division of Oncogenomics, Oncode Institute, Netherlands Cancer Institute, Amsterdam, Netherlands; [4]Division of Pathology, Netherlands Cancer Institute, Amsterdam, Netherlands; [5]Core Facility Molecular Pathology & Biobanking, Netherlands Cancer Institute, Amsterdam, Netherlands; [6]Department of Medical Oncology, Netherlands Cancer Institute, Amsterdam, Netherlands; [7]Department of Bioinformatics, Semmelweis University, Budapest, Hungary; [8]TTK Cancer Biomarker Research Group, Institute of Enzymology, Budapest, Hungary; [9]Department of Pediatrics, Semmelweis University, Budapest, Hungary; [10]Laboratory of Chemical Biology and Institute for Complex Molecular Systems, Department of Biomedical Engineering, Eindhoven University of Technology, Eindhoven, Netherlands

*For correspondence:
s.palit@nki.nl (SALP);
ms.vd.heijden@nki.nl (MSH)

**Abstract** Androgen receptor (AR) inhibitors represent the mainstay of prostate cancer treatment. In a genome-wide CRISPR-Cas9 screen using LNCaP prostate cancer cells, loss of co-repressor *TLE3* conferred resistance to AR antagonists apalutamide and enzalutamide. Genes differentially expressed upon *TLE3* loss share AR as the top transcriptional regulator, and *TLE3* loss rescued the expression of a subset of androgen-responsive genes upon enzalutamide treatment. GR expression was strongly upregulated upon AR inhibition in a *TLE3*-negative background. This was consistent with binding of TLE3 and AR at the *GR* locus. Furthermore, GR binding was observed proximal to TLE3/AR-shared genes. GR inhibition resensitized *TLE3*^KO cells to enzalutamide. Analyses of patient samples revealed an association between TLE3 and GR levels that reflected our findings in LNCaP cells, of which the clinical relevance is yet to be determined. Together, our findings reveal a mechanistic link between TLE3 and GR-mediated resistance to AR inhibitors in human prostate cancer.

## Introduction

Prostate cancer is the second most common cancer and the fifth leading cause of cancer-related death in men worldwide (*Torre et al., 2015*). Deregulated androgen receptor (AR) signaling is a major driver of prostate cancer (*Taylor et al., 2010*). Consequently, androgen deprivation therapy (ADT) is used to treat locally advanced and metastatic prostate cancer, achieving remission in most patients. However, despite castrate-levels of androgens in the serum, the disease inevitably progresses to a castration-resistant state (*Perlmutter and Lepor, 2007*). AR signaling remains a pivotal driver in castration-resistant prostate cancer (CRPC), which is illustrated by the efficacy of AR-

directed drugs such as abiraterone and enzalutamide. Unfortunately, patients develop resistance to these drugs and invariably succumb to the disease (*Clegg et al., 2012*; *Beer et al., 2014*; *Chi et al., 2019*).

Several resistance mechanisms to AR inhibitors have been proposed, including mutations in *AR* (*Korpal et al., 2013*; *Joseph et al., 2013*; *Prekovic et al., 2016*; *Prekovic et al., 2018*) and expression of splice variants (*Li et al., 2013*; *Antonarakis et al., 2014*; *Culig, 2017*). For example, the F877L missense mutation in *AR* was shown to confer resistance to enzalutamide and apalutamide (*Korpal et al., 2013*; *Joseph et al., 2013*; *Balbas et al., 2013*). Upregulation of the glucocorticoid receptor (GR, gene symbol *NR3C1*) was shown to be associated with clinical resistance to enzalutamide (*Arora et al., 2013*). Using the preclinical model LREX (LNCaP/AR Resistant to Enzalutamide Xenograft derived), it was shown that AR and GR have overlapping cistromes and transcriptomes, allowing GR to drive enzalutamide-resistant growth by regulating expression of a subset of AR target genes. (*Arora et al., 2013*; *Shah et al., 2017*). Significant overlap between AR and GR cistromes and transcription programs in prostate cancer cells was also described by others (*Sahu et al., 2013*). GR upregulation was found to occur through abrogation of the repressive function of AR and EZH2-mediated methylation of the *GR* enhancer (*Shah et al., 2017*). How exactly GR deregulation is mediated is incompletely understood. Combined, these studies have provided valuable insights into the molecular mechanisms underlying enzalutamide resistance in prostate cancer. However, to the best of our knowledge, a genome-scale approach aimed at identifying novel regulators of AR inhibitor sensitivity has hitherto not been reported. Loss-of-function genetic screens facilitate the unbiased identification of genes that have a central role in biological processes in various genetic and pharmacological backgrounds. Consequently, large-scale gene perturbation experiments are a powerful tool to identify novel drug targets and biomarkers of drug response (*Mullenders and Bernards, 2009*). Using this technology, we aimed to discover genes not previously implicated in enzalutamide resistance. Through a genome-wide CRISPR-Cas9 screen we identified transducin-like enhancer of split 3 (*TLE3*) as a modulator of AR inhibitor sensitivity that, upon loss, confers resistance to enzalutamide in prostate cancer cells.

The well-conserved TLE protein family of transcriptional co-repressors is expressed in the nucleus of metazoans and regulate various biological processes including development, cell metabolism, growth and differentiation. At the chromatin, TLE protein family members maintain a silenced chromatin structure (*Agarwal et al., 2015*; *Chen and Courey, 2000*; *Cinnamon and Paroush, 2008*). TLE3 is deregulated in various cancers including hormone-driven breast cancer (*Jangal et al., 2014*), colorectal cancer (*Yang et al., 2016*) and prostate cancer (*Nakaya et al., 2007*). Here, we report an unexpected role for TLE3 in regulating AR-mediated repression of the *GR* locus affecting AR inhibitor sensitivity in prostate cancer cells.

## Results

### A genome-wide CRISPR-Cas9 resistance screen identifies TLE3 as a novel regulator of AR inhibitor sensitivity

The androgen-dependent prostate cancer cell line LNCaP is sensitive to AR inhibitors such as apalutamide (*Figure 1—figure supplement 1A*) and enzalutamide (*Figure 1—figure supplement 1B*), making it a model system well-suited for the unbiased discovery of novel regulators of AR inhibitor sensitivity in prostate cancer cells. LNCaP cells were infected with a lentiviral pool containing the genome-wide scale CRISPR Knock-Out (GeCKO) half-library A (*Sanjana et al., 2014*), targeting 19052 genes with 3 gRNAs per gene. Infected cells were cultured in the presence of vehicle or 2 μM of the AR inhibitor apalutamide for 6 weeks to allow selection of resistant cells. Subsequently, barcodes were recovered from the cells and submitted for massively parallel sequencing (*Figure 1A* and *Figure 1—source data 1*). DESeq2 (*Love et al., 2014*) analysis (*Figure 1—source data 2*) and MAGeCK (*Li et al., 2014*) analysis (*Figure 1—source data 3*) both identified *TLE3* as the top hit with all three gRNAs enriched in cells treated with apalutamide compared to untreated cells (*Figure 1B* and *Figure 1—figure supplement 1C*).

The screen was performed using apalutamide (*SPARTAN Investigators et al., 2018*), which is a next-generation AR inhibitor structurally similar to enzalutamide. Subsequently, we validated the screen hit *TLE3* using both compounds. Abrogation of *TLE3* expression using independent single

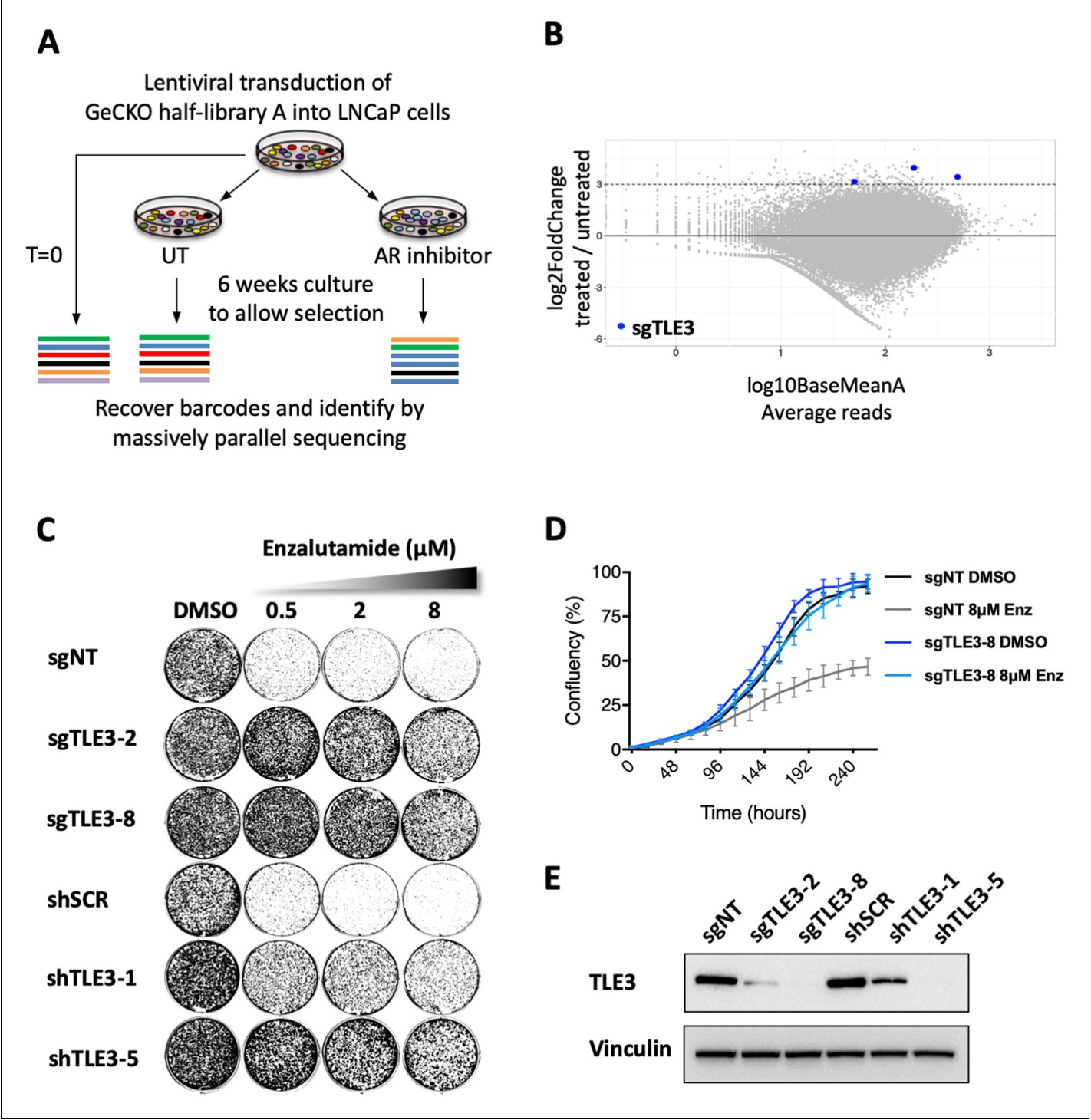

**Figure 1.** Genome-wide screen identifies TLE3 as a modulator of AR inhibitor sensitivity. (**A**) Overview of the genome-wide CRISPR-Cas9 resistance screen. (**B**) Representation of the relative abundance of the gRNA barcode sequences of the CRISPR-Cas9 resistance screen. The y-axis shows the enrichment (relative abundance of apalutamide treated/untreated) and the x-axis shows the average sequence reads of the untreated samples. (**C**) Long-term growth assay (14 days) showing the functional phenotype of LNCaP cells harboring *TLE3* knockout or knockdown vectors, cultured in the presence of vehicle or enzalutamide. Cells harboring a non-targeting sgRNA (sgNT) or scrambled shRNA (shSCR) were used as a control. (**D**) Quantitative analysis of live cell proliferation in real-time for control cells and *TLE3*[KO] cells in the absence or presence of enzalutamide. (**E**) Western blot showing TLE3 protein levels for control cells and *TLE3*[KO] cells shown in *C* and *D*. Vinculin was used as a loading control.

The online version of this article includes the following source data and figure supplement(s) for figure 1:

**Source data 1.** Normalized readcounts CRISPR screen.

*Figure 1 continued on next page*

*Figure 1 continued*
**Source data 2.** DESeq2 analysis of the CRISPR screen.
**Source data 3.** MAGeCK analysis of the CRISPR screen.
**Figure supplement 1.** Genome-wide CRISPR screen identifies TLE3 as a modulator of AR inhibitor sensitivity.

guide RNAs (sgRNAs), as well as short hairpin RNAs (shRNAs) targeting *TLE3*, conferred resistance to both enzalutamide and apalutamide in LNCaP cells in long-term growth assays (14 days) with drug concentrations up to 8 µM (*Figure 1C–E* and *Figure 1—figure supplement 1D and E*). Because enzalutamide is the current standard used in the clinic for the treatment of castration-resistant prostate cancer (CRPC), we used this drug for subsequent experiments. As prostate cancer is considered a heterogenous disease, for which only a few cell lines are available of which a subset is AR-driven, we next tested whether drug resistance as a result of *TLE3* loss could be confirmed in two other prostate cancer cell lines; CWR-R1 and LAPC4. As *TLE3* loss did not confer drug resistance in these two cell lines (*Figure 1—figure supplement 1F–H*), we conclude a context-dependency of this mode of resistance that is not commonly observed in all model systems.

## Loss of TLE3 leads to persistent expression of a subset of androgen-responsive genes in the presence of enzalutamide

TLE3 is known to be a negative regulator of the Wnt pathway. Analysis of active ß-catenin levels and expression of the *bona fide* Wnt target gene *AXIN2* in *TLE3*KO cells treated with vehicle or enzalutamide revealed no changes compared to control cells (*Figure 2—figure supplement 1A and B*), indicating Wnt signaling is not altered in this context.

To investigate the transcriptional consequences of *TLE3* abrogation in LNCaP prostate cancer cells, the transcriptomes of control and *TLE3*KO cells were compared (*Figure 2—source data 1* and GSE130246). Because TLE3 is a transcription co-factor, we analyzed differentially expressed genes for transcription factor enrichment to explore which pathways could be involved in enzalutamide resistance conferred by *TLE3* loss. Enrichment analysis revealed AR as the top transcription factor associated with genes differentially expressed in control cells versus *TLE3*KO cells in vehicle condition (*Figure 2—figure supplement 1C*). Genes differentially expressed in control cells versus *TLE3*KO cells, cultured in the presence of enzalutamide, also shared AR as the top regulator (*Figure 2A*). An overview of the most differentially expressed genes in control cells versus *TLE3*KO cells treated with enzalutamide is shown in *Figure 2B*. We validated expression for several of these genes both in the absence and presence of enzalutamide and found that loss of *TLE3* rescued expression of these genes in cells exposed to enzalutamide (*Figure 2C* and *Figure 2—figure supplement 1D*). We next asked the question whether general AR signaling is restored upon *TLE3* loss in enzalutamide-treated cells. To test this, we performed gene set enrichment analysis (GSEA) selectively focusing on AR-responsive genesets. Overall, AR signaling was maintained in the presence of enzalutamide in *TLE3*KO cells but not in control cells, implying a rescue of AR signaling despite enzalutamide treatment (*Figure 2D and E* and *Figure 2—figure supplement 1E*).

Based on its role in the regulation of AR target genes and AR inhibitor resistance, we hypothesized that *TLE3* itself may be androgen-regulated. Indeed, western blot analysis showed that in wild-type (WT) cells, the expression of TLE3 is induced by enzalutamide (*Figure 2—figure supplement 1F*). Conversely, stimulation with the synthetic androgen R1881 led to a decrease in TLE3 protein levels (*Figure 2—figure supplement 1F*). Hormone manipulation led to similar changes in LAPC4 and CWR-R1 cells, although to a much lesser extent (*Figure 2—figure supplement 1G*). Analysis of publicly available ChIP-seq data (*Stelloo et al., 2018*) revealed binding of both TLE3 and AR at enhancer sites of the *TLE3* gene (*Figure 2—figure supplement 1H*) suggesting that these transcription factors regulate *TLE3* expression, indicating a feedback loop controlling *TLE3* transcription. This is supported by analysis of publicly available RNA-seq data (*Massie et al., 2011*) showing that *TLE3* mRNA levels are downregulated over time in LNCaP cells that are treated with R1881 (*Figure 2—figure supplement 1I*). Finally, we also investigated the effect of *TLE3* loss on *AR* gene expression using qPCR and found no differential expression for *AR* mRNA between control and *TLE3*KO cells (*Figure 2—figure supplement 1J*).

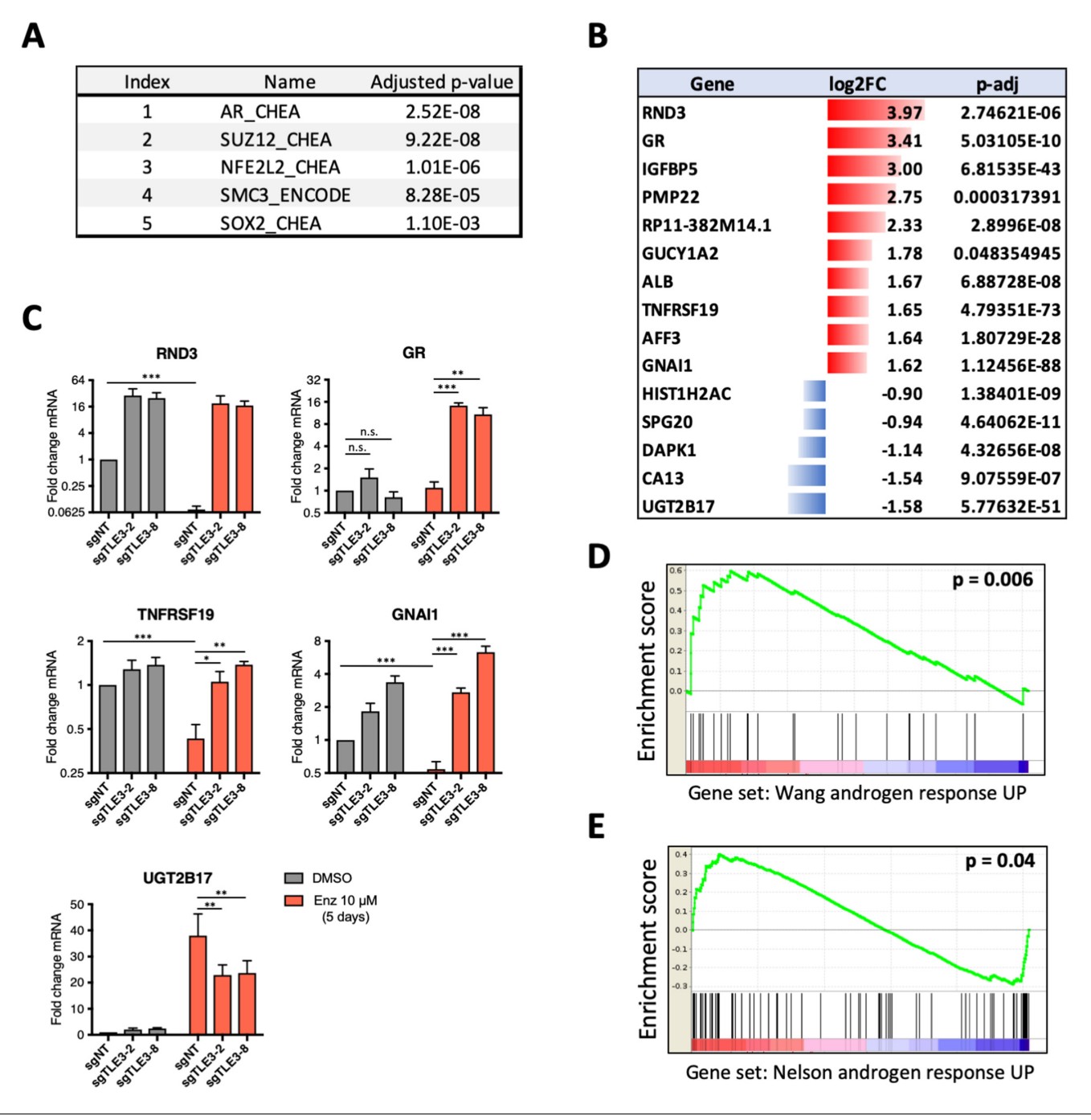

**Figure 2.** Transcriptomics analyses comparing control and *TLE3*[KO] cells cultured in the presence of vehicle or 10 μM enzalutamide for 5 days. (A) Enrichment analysis for transcription factors associated with genes differentially expressed in enzalutamide-treated control cells compared to *TLE3*[KO] cells. (B) Overview of the fold changes in gene expression of the most differentially expressed genes in control cells versus *TLE3*[KO] cells treated with enzalutamide. (C) Validation (qPCR) of mRNA expression levels for several genes shown in *B*. Bars represent average data from at least three independent experiments ± SEM. P-values are indicated with \*\*\*p<0.001, \*\*p<0.01 and \*p<0.05 (two-tailed *t*-test). (D–E) GSEA for genes differentially expressed in control cells compared to *TLE3*[KO] cells, treated with 10 μM enzalutamide using indicated gene sets.

The online version of this article includes the following source data and figure supplement(s) for figure 2:

**Source data 1.** Readcounts RNA-seq experiment comparing control and *TLE3*[KO] cells.
**Figure supplement 1.** Transcriptomics analyses comparing control and *TLE3*[KO] cells.

## TLE3 localizes at AR binding sites proximal to genes differentially expressed in *TLE3*[KO] cells compared to control cells

Gene expression profiling of *TLE3*[KO] cells revealed persistent expression of androgen-responsive genes in the presence of enzalutamide. Recently, protein interactome profiling of AR revealed that TLE3 binds together with FOXA1 at androgen response elements (AREs) (*Stelloo et al., 2018*). We next analyzed publicly available ChIP-seq data (*Stelloo et al., 2018*) for the genome-wide binding profiles of AR, TLE3 and FOXA1 in LNCaP cells to explore the role of these transcription factors in the direct regulation of genes differentially expressed in control cells compared to *TLE3*[KO] cells under enzalutamide treatment. Genes showing the strongest log2 fold-change expression in *TLE3*[KO] compared to control cells were indeed bound by TLE3 (*Figure 3—figure supplement 1A*). In *Figure 3A*, the coverage profiles for TLE3 and AR are shown at the loci of two genes (*RND3* and *GNAI1*) whose expression was found to correlate with *TLE3*[KO] and enzalutamide treatment (*Figure 3A* and *Figure 2B and C*). Genome-wide analysis of the binding patterns for AR, TLE3 and

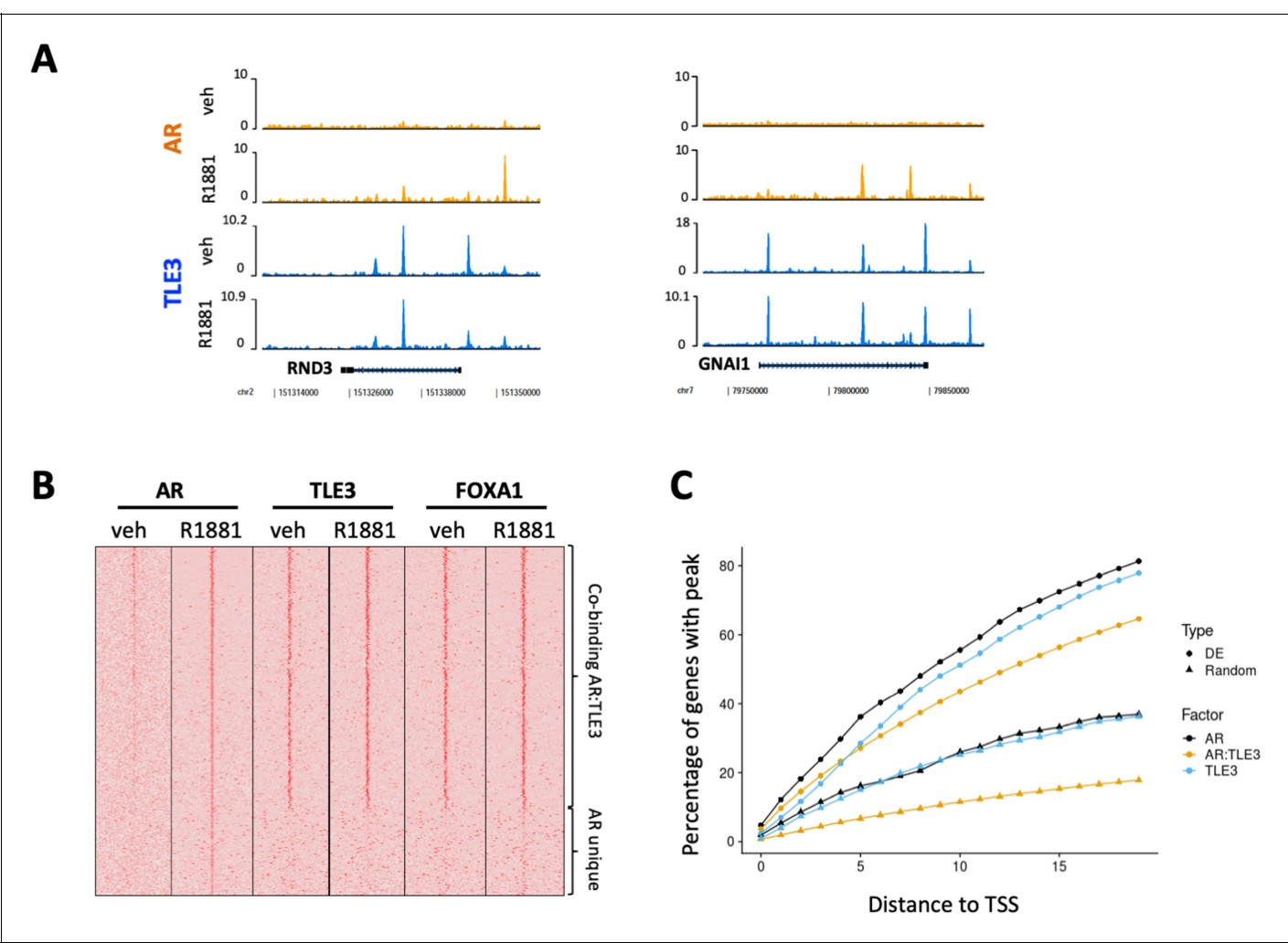

**Figure 3.** ChIP-seq analyses for transcription factor binding at differentially expressed genes in control cells compared to *TLE3*[KO] cells cultured in the presence of 10 µM enzalutamide. (**A**) Coverage profiles for TLE3 and AR at the loci of two genes (*RND3* and *GNAI1*). (**B**) Heatmap of AR, TLE3 and FOXA1. (co-)binding at genes differentially expressed in *TLE3*[KO] compared to control cells treated with enzalutamide are shown. The binding of AR, TLE3 and FOXA1 at these sites is shown for androgen-depleted or R1881-stimulated (4 hr) conditions in parental LNCaP cells. (**C**) ChIP-seq peak enrichment near the Transcription Start Sites (TSS) of differentially expressed (DE) genes and a random set of genes. The fraction of genes with a peak for TLE3, AR or both transcription factors at indicated distance from the TSS is shown for both genesets.

The online version of this article includes the following figure supplement(s) for figure 3:

**Figure supplement 1.** TLE3 binding status proximal to genes differentially expressed control cells compared to *TLE3*[KO] cells.

FOXA1 at the regulatory elements of differentially expressed genes extended our findings more broadly showing overlap for these proteins at these sites with markedly similar binding profiles observed for TLE3 and FOXA1 (*Figure 3B*). We found that co-binding of TLE3 and AR was enriched at loci of the differentially expressed geneset when compared to a random geneset (*Figure 3C*). Furthermore, significantly enriched sequence motifs at TLE3 binding sites of differentially expressed genes included members of the forkhead box transcription factor family (including FOXA1), AR, HOXB13 and the glucocorticoid receptor (GR) (*Supplementary file 1*). Since TLE3 acts as a repressor, the chromatin binding profiles for TLE3, FOXA1 and AR substantiate the expression data indicating that loss of *TLE3* alters expression of androgen-responsive genes towards an active-AR-like profile in spite of anti-hormonal treatment, thereby allowing continued growth when these cells are exposed to enzalutamide.

## Enzalutamide resistance in *TLE3*^KO cells occurs through GR which is upregulated upon AR inhibition

Gene expression analysis revealed that *GR* was one of the most upregulated genes upon enzalutamide treatment in a *TLE3*-loss background (*Figure 2B and C*). Western blot analysis for GR confirmed this upregulation on protein level (*Figure 4A*). The binding of TLE3 and AR at the *GR* enhancer provides further evidence that both proteins play a role in the transcriptional repression of *GR* (*Figure 4B*). Moreover, TLE3 and AR binding at this region occurs at the same regulatory element described previously to be relevant in the regulation of *GR* in prostate cancer progression (*Shah et al., 2017*) (*Figure 4—figure supplement 1A*). The core GR and AR consensus sequences are highly similar (*Figure 4—figure supplement 1B*), and GR sequence motifs were enriched at genes differentially expressed in control versus *TLE3*^KO cells cultured with enzalutamide (*Supplementary file 1*). Interestingly, GR has been implicated in mediating resistance to AR inhibitors (*Arora et al., 2013*; *Shah et al., 2017*) so we decided to further investigate the link between TLE3 and GR in the context of antihormonal therapy. To assess whether GR can act as a key effector in *TLE3*^KO cells resulting in drug resistance, we performed inhibition experiments for this receptor in the context of enzalutamide treatment comparing control and *TLE3*^KO cells. Inhibition of GR using shRNAs in control and *TLE3*^KO cells resensitized *TLE3*^KO cells to enzalutamide (*Figure 4C–E* and *Figure 4—figure supplement 1C*). Inhibition of GR using the small molecule inhibitor mifepristone in conjunction with enzalutamide, reduced the proliferation of *TLE3*^KO when compared to single-drug treatments (*Figure 4F*). We next performed ChIP-qPCR to determine GR chromatin binding proximal to several of the most-differentially expressed genes in control versus *TLE3*^KO cells, in the presence of enzalutamide (listed in *Figure 2B*). This experiment showed binding of GR at these loci only in *TLE3*^KO cells treated with enzalutamide (*Figure 4G*). As TLE3 is known to recruit HDACs (*Chen and Courey, 2000*; *Cinnamon and Paroush, 2008*), we also investigated histone acetylation at the GR locus and AR/TLE3 target gene RND3 and found that loss of TLE3 resulted in an upregulation of H3K27 acetylation at these enhancers (*Figure 4—figure supplement 1D*). Thus, abrogation of the repressive function mediated by both AR and TLE3 at the *GR* locus allows for increased expression of GR which, in turn, is able to confer enzalutamide resistance by substituting for AR in this context. Interestingly, we found overlap between several of the most-differentially expressed genes listed in *Figure 2* (RND3, GNAI1, GR, UGT2B17 and PMP22) and GR-regulated genes described in a model for GR-mediated enzalutamide resistance as reported by others (*Arora et al., 2013*) (*Figure 4—figure supplement 1E*). These results are further supported by previous findings showing that AR and GR have overlapping transcriptomes and cistromes in the LNCaP-derived enzalutamide-resistant cell model LREX where GR was shown to confer enzalutamide resistance (*Arora et al., 2013*; *Shah et al., 2017*). Together, our data shows that loss of *TLE3* in conjunction with AR inhibition results in GR upregulation, leading to enzalutamide-resistance in LNCaP prostate cancer cells.

## TLE3 and GR expression are inversely correlated in prostate cancer patients and TLE3^low/GR^high expression is associated with poor response to antihormonal therapy

Analysis of two publicly available RNA-seq datasets (TCGA prostate and *Abida et al., 2019* ) revealed an inverse correlation between TLE3 expression and GR expression in biopsy samples from

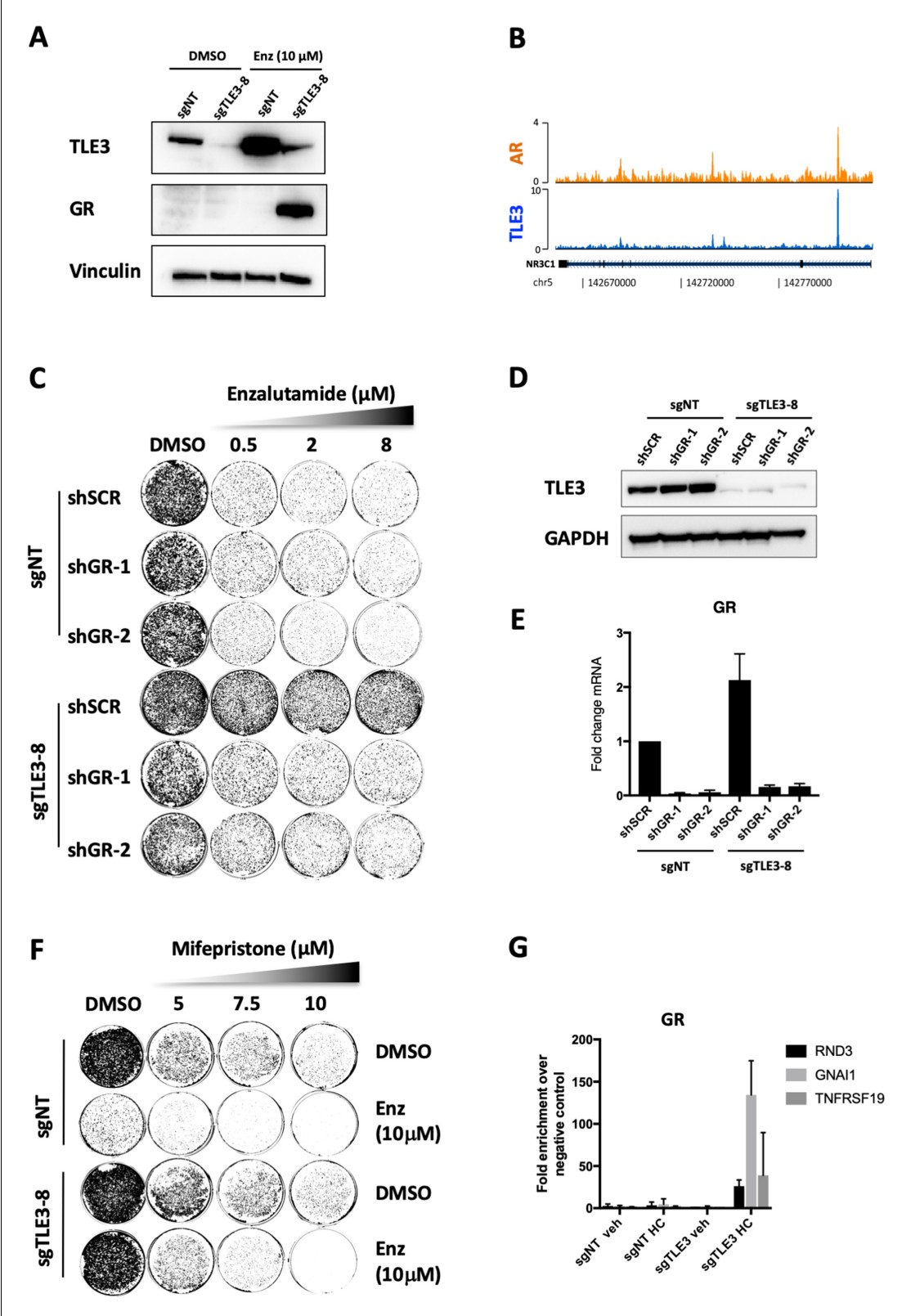

**Figure 4.** GR inhibition resensitizes *TLE3*^KO cells to enzalutamide treatment. (**A**) Western blot showing protein expression levels of TLE3 and GR in control and *TLE3*^KO cells cultured vehicle or enzalutamide (**B**) Coverage profiles for TLE3 and AR binding at the *GR* locus. (**C**) Long-term growth assay (14 days) showing the drug resistance phenotype of control and *TLE3*^KO cells with and without *GR* knockdown in the presence of vehicle or enzalutamide. (**D**) Western blot analysis for TLE3 protein levels in control and *TLE3*^KO cells shown in *C*, using GAPDH as a loading control. (**E**) mRNA
*Figure 4 continued on next page*

*Figure 4 continued*

levels for *GR* in control and *TLE3*<sup>KO</sup> cells carrying shSCR or shGR constructs, shown in *C*. (**F**) Long-term growth assay (14 days) for cells harboring a control sgRNA or *TLE3*-targeting sgRNA cultured in the presence of vehicle, enzalutamide, mifepristone or the combination at indicated concentrations. (**G**) ChIP-qPCR showing GR occupancy at enhancers proximal to indicated genes. All samples were cultured in the presence of 10 µM enzalutamide with or without 1 µg/ml hydrocortisone (HC) as indicated.

The online version of this article includes the following figure supplement(s) for figure 4:

**Figure supplement 1.** GR-mediated gene regulation in *TLE3*<sup>KO</sup> cells.

prostate cancer patients with early-stage disease (*Figure 5A*) as well as advanced prostate cancer (*Figure 5B*).

We next investigated the effect of TLE3 expression levels on disease progression in prostate cancer patients. Analysis of the TCGA prostate cancer patient dataset filtered for patients who had undergone anti-hormonal therapy revealed a correlation between TLE3 expression and biochemical recurrence (p = 0.033, n = 65) (*Figure 5C*). These data show that TLE3 expression is a prognostic factor for prostate cancer patients treated with anti-hormonal therapy.

As part of a clinical trial run in-house (PRESTO), matched tissue samples of metastatic sites were collected before treatment and after progression on enzalutamide treatment for four CRPC patients. These paired biopsies were analyzed by immunohistochemistry for expression of TLE3 and GR, to investigate whether expression of these proteins is altered upon selection pressure by enzalutamide. Two patients had a short PSA response to enzalutamide (<6 months), without radiological response. Tumor tissue of these patients showed moderate to high GR expression at baseline with weak or negative staining for TLE3 (*Figure 5D*, and *Figure 5—figure supplement 1A*). This was also observed in the post-treatment samples from these patients, in agreement with our hypothesis of low TLE3 and high GR in resistant tumors. Moreover, for one of these patients, the inverse association between TLE3 and GR became more pronounced upon enzalutamide treatment (*Figure 5D*). The third patient, having a more profound response (PSA response >12 months, radiological response), had weak staining for TLE3 and moderate staining for GR at baseline. In the post-treatment staining, TLE3 was low, whereas GR expression had increased (*Figure 5D*). The fourth patient had a protracted response to enzalutamide (>2 years) and showed low expression of both TLE3 and GR in pre- and post-treatment tissue (*Figure 5—figure supplement 1A*). In this patient, amplification of AR was observed upon treatment, potentially explaining resistance not related to TLE3 expression. Combined, these data show that TLE3 and GR are inversely correlated in prostate cancer patient samples and that low TLE3 and high GR expression were observed in several cases of enzalutamide resistance.

## Discussion

The efficacy of the AR antagonists enzalutamide and apalutamide illustrates the importance of persistent signaling through the AR pathway in CRPC (*Clegg et al., 2012*; *Beer et al., 2014*). The transient nature of these drug responses underscores the relevance of improving therapeutic approaches and mechanistic understanding of drug resistance (*Prekovic et al., 2018*). Using a genome-wide CRISPR-Cas9 resistance screen we identified TLE3 as a novel regulator of AR inhibitor sensitivity that binds to and regulates the expression of androgen-responsive genes.

TLE3 was shown to co-localize with FOXA1 and AR at enhancer elements, which are selectively activated during prostate tumorigenesis (*Stelloo et al., 2018*), underscoring the importance of these transcription factors in this context. Our gene expression analyses show that loss of *TLE3* results in an active-AR-like profile despite anti-hormonal treatment. Our findings are in line with TLE3's known role as a transcriptional repressor (*Agarwal et al., 2015*; *Chen and Courey, 2000*; *Cinnamon and Paroush, 2008*) and the fact that TLE3 binds AR target genes. Similarly, TLE3 was described as a co-repressor in breast cancer cells, where it co-regulates the expression of a subset of ERα target genes (*Jangal et al., 2014*). The same study showed that the binding of TLE3 to the chromatin at ERα target genes was dependent on FOXA1 (*Jangal et al., 2014*).

Pathway reactivation or feedback activation of parallel signaling pathways are commonly described mechanisms found in drug-resistant tumors treated with targeted therapy (*Prahallad et al., 2012*; *Pawar et al., 2018*). In enzalutamide-resistant prostate tumors, upregulation

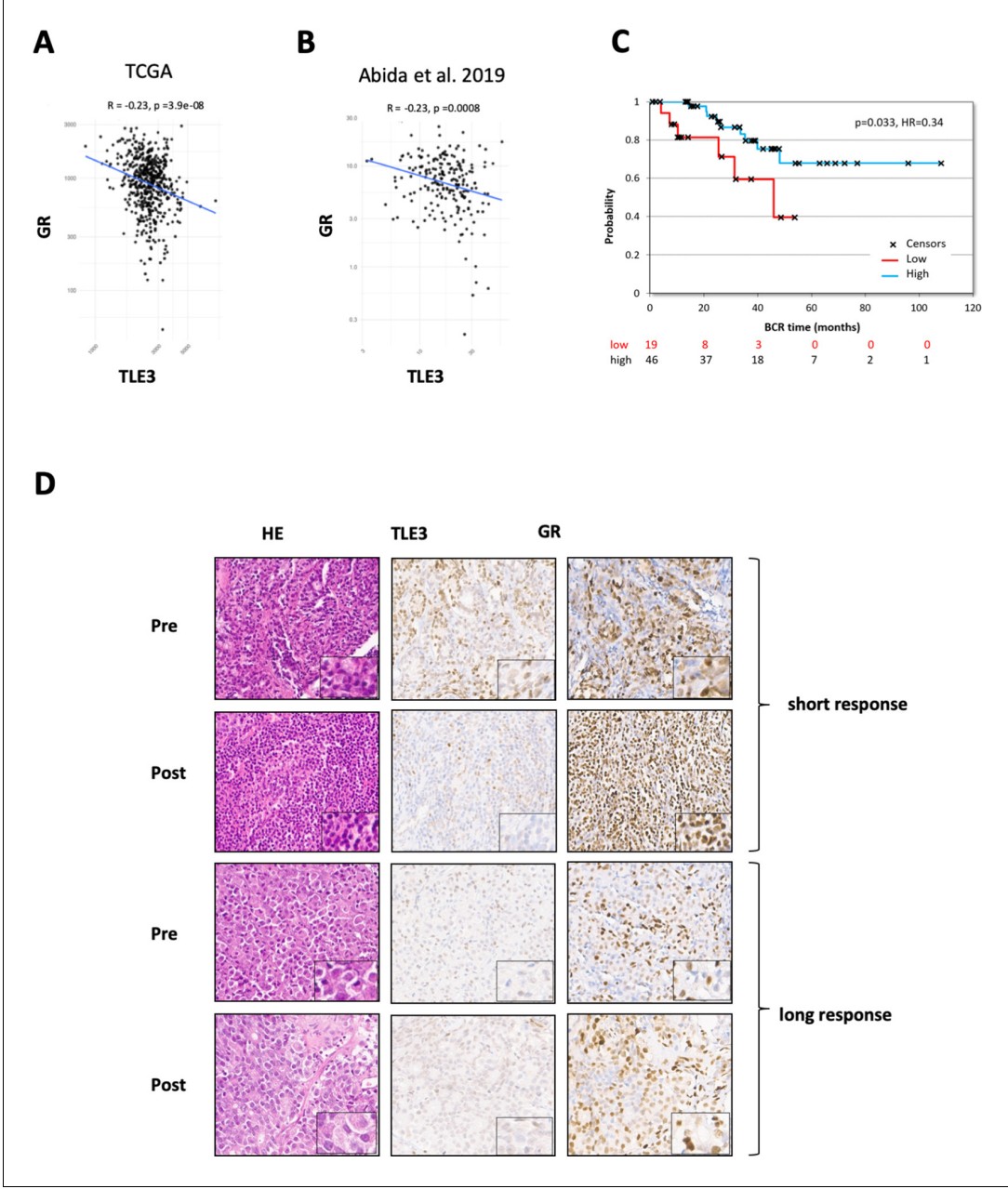

**Figure 5.** TLE3 and GR expression in tumors of prostate cancer patients. (A–B) RNA-seq analysis showing the correlation between TLE3 and GR expression in tumor samples from prostate cancer patients. (C) Kaplan-Meier curve showing the biochemical recurrence of prostate cancer patients from the TCGA dataset, only patients receiving anti-hormonal therapy were included (65 patients) using an optimal cut-off for high versus low TLE3 expression. (D) Immunohistochemistry for H and E, TLE3 and GR in tumor biopsy samples collected from two CRPC patients pre- and post-enzalutamide treatment.

The online version of this article includes the following figure supplement(s) for figure 5:

**Figure supplement 1.** TLE3 and GR protein expression in CRPC patient samples pre- and post enzalutamide treatment.

of GR was described as a resistance mechanism where the receptor was able to substitute for AR and drive expression of a subset of target genes (*Arora et al., 2013*). In this study, the GR upregulation observed in the preclinical LREX model was not immediate in response to enzalutamide but required treatment with the drug for an extended amount of time for adaptation in vitro (*Arora et al., 2013*). This extended period of time needed for adaptation could suggest an acquired loss of TLE3 expression over time, resulting in deregulated GR expression. The work of *Shah et al. (2017)* showed that loss of the repressive signals of both AR binding and EZH2-mediated methylation of a tissue-specific enhancer at the GR locus lead to upregulation of GR and drug resistance in prostate cancer cells. TLE3 is a known transcriptional repressor and is able to bind the same GR enhancer (*Figure 4B* and *Figure 4—figure supplement 1A*). Our finding that TLE3 loss, in conjunction with AR inhibition, leads to GR upregulation provides deeper insight into the epigenetic regulation of the GR locus in prostate cancer cells and supports the previously undescribed role of TLE3 in conferring enzalutamide sensitivity via GR. GR occupancy at several TLE3/AR target genes provides further evidence for the role of GR in mediating enzalutamide resistance. Importantly, several of the most differentially expressed target genes in enzalutamide-treated control cells compared to *TLE3*^KO cells (RND3, GNAI1, GR, UGT2B17 and PMP22) were previously described to be GR-regulated in a model of AR inhibitor resistance. Together, our results provide novel insights into the regulation of the *GR* locus in the context of AR inhibition in prostate cancer cells, implicating TLE3 as a regulator of GR-mediated enzalutamide resistance.

A limitation of our study is the fact that, of the in vitro models we tested, loss of *TLE3* conferred resistance to enzalutamide only in LNCaP cells and not in LAPC4 and CWR-R1. The availability of in vitro prostate cancer models is limited, and the heterogeneous nature of resistance mechanisms to antihormonal therapies in prostate cancer may explain why *TLE3* loss did not confer resistance to enzalutamide in LAPC4 and CWR-R1 cells. To study broader applicability, we investigated several clinical data-sets. Analysis of RNA expression in two prostate cancer patient cohorts, showed an inverse correlation between TLE3 and GR expression and worse prognosis of prostate cancer patients with low TLE3 expression treated with antihormonal therapy. Additionally, immunohistochemistry on GR and TLE3 of tumor tissue collected from CRPC patients pre- and post-enzalutamide treatment support our findings in LNCaP cells. Although these observations are in agreement with our hypothesis, the clinical implications of our findings are yet to be resolved and need to be determined in larger cohorts. Thus, our results warrant further investigation into the role of TLE3 and enzalutamide resistance in prostate cancer patients.

In summary, we have identified TLE3 loss as a novel resistance mechanism to AR-targeted therapeutics in prostate cancer cells. Based on previously reported work and the data in our study, we propose a model in which loss of TLE3 and AR function at the GR enhancer leads to upregulation of GR, which is able to substitute for AR, resulting in enzalutamide resistance (*Figure 6*). Our data, implicating TLE3 in the regulation of *GR* expression and drug resistance, complements increasing evidence describing the role of this receptor in bypassing AR blockade in prostate cancer cells.

## Materials and methods

For an overview of the resources used for this study see *Supplementary file 2*: Key Resources Table.

### Cell culture and generation of knockout and knockdown cells

The human prostate cancer cell lines were maintained in RPMI (LNCaP, CWR-R1, 22rv1) or IMDM (LAPC4). HEK293T cells were cultured in DMEM. Medium was supplemented with 10% FBS and 1% penicillin/streptomycin. Cells were maintained at 37 °C in 5% $CO_2$. All cell lines were STR profiled. Control and *TLE3*^KO cells were created by infecting target cells with lentiviral particles containing LentiCRISPR v2.0 harboring non-targeting or *TLE3*-targeting gRNAs, which were cloned in using Gibson Assembly (NEB cat#: E2611S) utilizing BsmBI restriction sites. For gRNA and shRNA sequences see *Supplementary file 2*: Key Resources Table. HEK293 were co-transfected with lentiviral CRISPR, or in-house shRNA constructs, using PEI. Target cells were seeded 1 day prior to infection. Lentiviral supernatant was added to the medium along with 5 µg/ml polybrene. Infected cells were selected with 2 µg/ml puromycin.

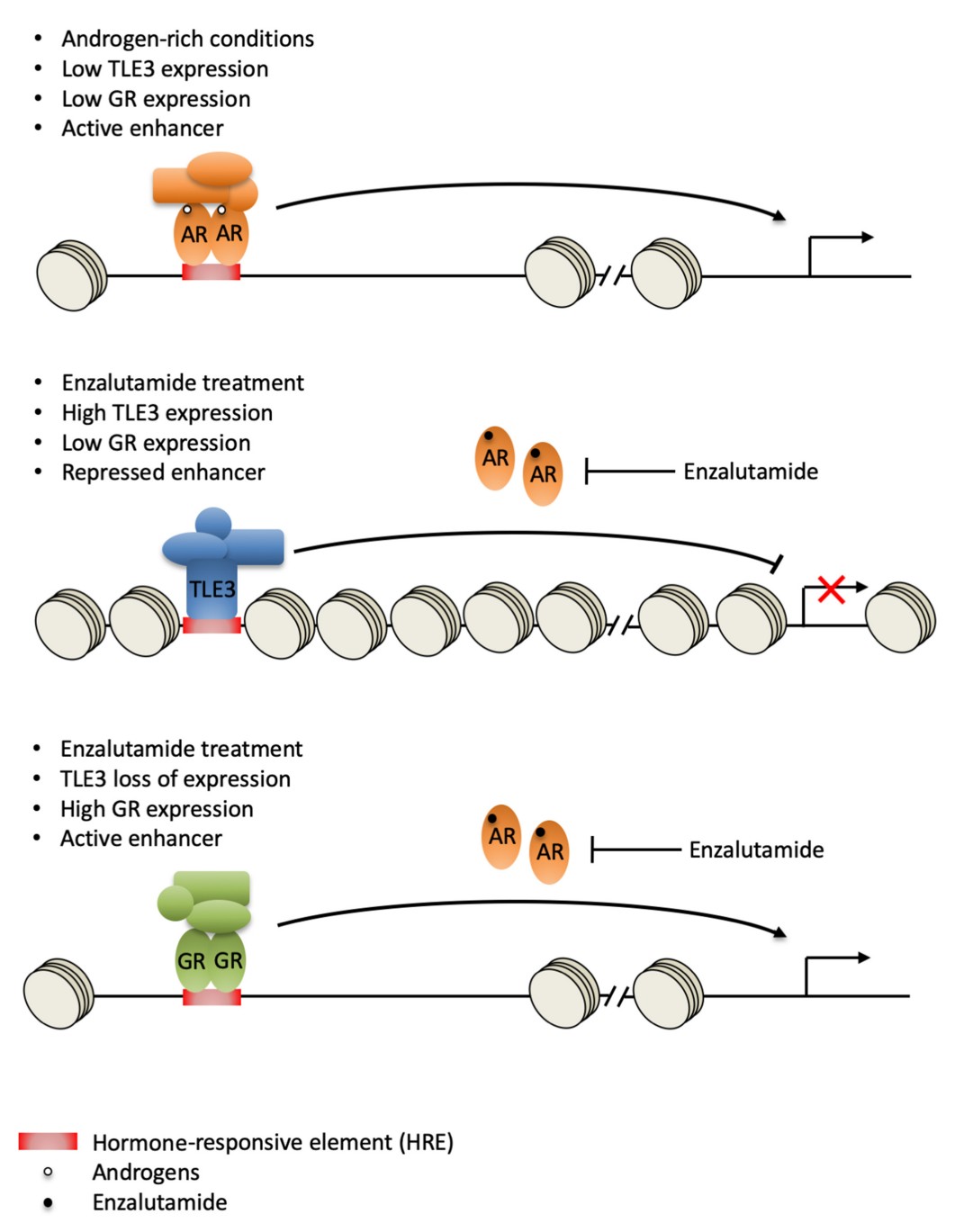

**Figure 6.** Model for GR-mediated enzalutamide resistance in *TLE3*^KO prostate cancer cells. In the presence of androgens, TLE3 expression is repressed and enhancers are active. AR regulates target gene expression, including repression of the *GR* locus (top panel). Upon enzalutamide treatment, TLE3 is upregulated and enhancers are inactive. TLE3 represses expression of AR target genes including *GR* (middle panel). Enzalutamide treatment in the context of *TLE3* loss leads to upregulation of GR which is able to substitute for AR at active enhancers, leading to drug resistance (bottom panel).

## CRISPR-Cas9 resistance screen

LNCaP cells were infected with lentiviral particles containing GeCKO half-library A at low M.O.I. (~0.2) for single viral integration, at a ~ 150 fold coverage, and cultured in the presence of vehicle or 2 μM apalutamide for 6 weeks. Barcodes were recovered and sequenced as described (*Brunen et al., 2018*). DESeq2 (*Love et al., 2014*) analysis was performed using a paired design. The treated samples were compared with the untreated samples. A sgRNA was considered to be a

hit, if the log2FC $\geq$ 3 and the FDR $\leq$ 0.1. TLE3 was the only gene for which all three sgRNAs were a hit. The MAGeCK (*Li et al., 2014*) analysis was done using the default settings, which produced TLE3 as top hit with a FDR of 0.002.

## Proliferation assays

Colony formation assays were performed as previously described (*Brunen et al., 2018*). Used seeding densities were 20,000 (LNCaP, LAPC4) or 10,000 (22rv1, CWR-R1) cells/well in 6-well plates. After 12–14 days of growth in presence of the drugs as indicated, when control cells reached confluence, all cells were fixed in 2% formaldehyde and stained with 0.1% crystal violet.

Live cell proliferation was monitored in real-time using the Incucyte ZOOM (11 days). Cells were seeded in a 384-well plate at 600 cells/well and drugs were added as indicated.

R1881, apalutamide, enzalutamide, mifepristone (Medkoo Biosciences) were dissolved in DMSO and stored at −20°C.

## Protein lysate preparation and western blot analysis

Typically, LNCaP cells were plated at density of 200,000 cells in 6-well plates and cultured in the presence of enzalutamide for 5 days before harvesting. Samples were prepared and western blot was performed as described previously (*Brunen et al., 2018*), using primary antibodies directed against TLE3 (Santa Cruz Biotechnology, #sc-514798, 1:250), Vinculin (Sigma-Aldrich, #V4139, 1:1000), and GAPDH (Cell Signaling Technology, #5174S, 1:10000). Secondary antibodies were obtained from Bio-Rad laboratories.

## RNA-seq

RNA-seq data was generated by seeding 500,000 LNCaP control or *TLE3*$^{KO}$ cells in 10 cm dishes in the presence of 10 μM enzalutamide or vehicle for 5 days, followed by RNA isolation using the ISO-LATE II RNA mini kit (Bioline). RNA was then submitted for Illumina sequencing (HiSeq 2500). The differential expression was based on the ratio of normalized read counts (FPKM, after library size correction). An absolute fold-change threshold of 2 was used. Genes with a coverage <50 in both conditions were excluded from the analyses to prevent spurious results. Data were further analyzed using Enrichr (*Chen et al., 2013*) and javaGSEA desktop application (http://software.broadinstitute.org/gsea). Data was uploaded to GEO (GSE130246).

## Quantitative RT-PCR

LNCaP cells were plated at density of 200,000 cells in 6-well plates and cultured in the presence of enzalutamide for 5 days before harvesting. Total mRNA isolation, cDNA synthesis and qPCR analysis were performed as described elsewhere (*Brunen et al., 2018*). An overview of the used primers is listed in *Supplementary file 2*: Key Resources Table.

## ChIP-seq and ChIP-qPCR

The ChIP-seq data from *Figure 3* was sourced from *Stelloo et al. (2018)*, GSEA94682. The sequencing (bam) files and the peaks called by Peaks were called using DFilter (*Kumar et al., 2013*) and MACS peak caller version 1.4 (*Zhang et al., 2008*). The ChIP peaks were sorted by intensity. For each set of differentially expressed genes, the genomic locations were intersected with the peaks called, padding with 20 Kb for genes and 5 Kb for peaks prior to intersecting. Seqminer (*Ye et al., 2011*) was used to obtain the coverage data at the intersecting regions, and to generate the heatmaps. Coverage profile snapshots were made using Easeq (*Lerdrup et al., 2016*).

ChIP-qPCR data was generated according to the protocol described by *Singh et al. (2019)*. Cells were plated at ~30% confluency in 15 cm dishes and cultured in the presence of 10 μM enzalutamide for 5 days. In case of hydrocortisone stimulation, hydrocortisone was added 2 hr prior to harvesting of the cells. The antibodies that were used were: 7,5 μl of anti-GR (CST, #12041) and 5 μg of H3K27ac (Active Motif, 39133). Regions for qPCR were selected based on AR ChIP-seq data in *Figure 3*, choosing the peaks closest the target gene. For an overview of the primers see *Supplementary file 2*: Key Resources Table.

## Immunohistochemistry

Immunohistochemistry of the FFPE tumor samples was performed on a BenchMark Ultra autostainer (TLE3) or Discovery Ultra autostainer (Glucocorticoid Receptor). Briefly, paraffin sections were cut at 3 μm, heated at 75°C for 28 min and deparaffinized in the instrument with EZ prep solution (Ventana Medical Systems). Heat-induced antigen retrieval was carried out using Cell Conditioning 1 (CC1, Ventana Medical Systems) for 64 min at 95°C.Glucocorticoid Receptor clone D6H2L (Cell Signaling) was detected using 1/600 dilution, 1 hr at 37°C and TLE3 using clone CL3573 (1/250 dilution, 1 hr at RT). Bound TLE3 was detected using the OptiView DAB Detection Kit (Ventana Medical Systems). Glucocorticoid Receptor bound antibody was visualized using Anti-Rabbit HQ (Ventana Medical systems) for 12 min at 37°C, Anti-HQ HRP (Ventana Medical systems) for 12 min at 37°C, followed by ChromoMap DAB Detection Kit (Ventana Medical Systems). Slides were counterstained with Hematoxylin and Bluing Reagent (Ventana Medical Systems).

## Acknowledgements

This work was funded by a KWF-Alpe d'HuZes grant (NKI 2014–7080). S Stelloo is funded by the Movember Foundation, and W Zwart is supported by a KWF-Alpe d'HuZes Bas Mulder Award and Netherlands Scientific Organization NWO VIDI grant. We would like to acknowledge Yanyun Zhu for technical support and helpful discussions. The authors thank the NKI Genomics Core Facility for bioinformatics support. We would like to acknowledge the NKI- AVL Core Facility Molecular Pathology and Biobanking (CFMPB) for supplying NKI-AVL Biobank material and lab support.

## Additional information

### Competing interests

Wilbert Zwart: Reviewing editor, *eLife*. The other authors declare that no competing interests exist.

### Funding

| Funder | Grant reference number | Author |
|---|---|---|
| KWF Kankerbestrijding | NKI2014-7080 | Andries M Bergman Wilbert Zwart Michiel S van der Heijden |
| Movember Foundation | | Suzan Stelloo |
| KWF Kankerbestrijding | Alpe d'HuZes Bas Mulder Award | Wilbert Zwart |
| NWO | VIDI grant | Wilbert Zwart |

The funders had no role in study design, data collection and interpretation, or the decision to submit the work for publication.

### Author contributions

Sander AL Palit, Conceptualization, Data curation, Formal analysis, Supervision, Validation, Investigation, Visualization, Methodology; Daniel Vis, Data curation, Software, Formal analysis, Validation, Investigation, Visualization, Methodology; Suzan Stelloo, Resources, Formal analysis, Investigation, Visualization; Cor Lieftink, Software, Formal analysis, Investigation; Stefan Prekovic, Elise Bekers, Resources, Formal analysis, Investigation; Ingrid Hofland, Resources, Investigation; Tonći Šuštić, Resources, Investigation, Methodology; Liesanne Wolters, Validation, Investigation; Roderick Beijersbergen, Resources, Formal analysis; Andries M Bergman, Resources, Funding acquisition; Balázs Győrffy, Resources, Data curation, Software, Formal analysis, Investigation, Visualization; Lodewyk FA Wessels, Resources, Software; Wilbert Zwart, Resources, Funding acquisition, Methodology; Michiel S van der Heijden, Conceptualization, Supervision, Funding acquisition, Methodology, Project administration

Author ORCIDs
Sander AL Palit (ID) https://orcid.org/0000-0003-2487-4311
Wilbert Zwart (ID) http://orcid.org/0000-0002-9823-7289

Decision letter and Author response
Decision letter https://doi.org/10.7554/eLife.47430.sa1
Author response https://doi.org/10.7554/eLife.47430.sa2

## Additional files

### Supplementary files

• Supplementary file 1. Motif enrichment analysis. Motif enrichment analysis showing sequence motifs that are enriched at genes differentially expressed in enzalutamide-treated control cells compared to *TLE3*$^{KO}$ cells.

• Supplementary file 2. Key Resources Table.

• Supplementary file 3. TCGA prostate cancer dataset (from https://portal.gdc.cancer.gov/) used for *Figure 5C*.

• Transparent reporting form

### Data availability

Data for Figure 1 (CRISPR resistance screen) is provided (source data file for Figure 1). Data for Figure 2 (RNA-seq) have been deposited in GEO under accession code GSE130246. Data (ChIP-seq) for Figure 3 and 4 is publicly available (GSE94682). Data for Figure 5C is provided (Supplementary file 3).

The following dataset was generated:

| Author(s) | Year | Dataset title | Dataset URL | Database and Identifier |
|---|---|---|---|---|
| Palit S, Vis D, Lieftink C | 2019 | RNA-seq control (sgNT) and TLE3KO (sgTLE3) cells treated with 10 uM enzalutamide or vehicle | http://www.ncbi.nlm.nih.gov/geo/query/acc.cgi?acc=GSE130246 | NCBI Gene Expression Omnibus, GSE130246 |

The following previously published dataset was used:

| Author(s) | Year | Dataset title | Dataset URL | Database and Identifier |
|---|---|---|---|---|
| Stelloo S, Zwart W | 2018 | Endogenous Androgen Receptor proteomic profiling reveals genomic subcomplex involved in prostate tumorigenesis | https://www.ncbi.nlm.nih.gov/geo/query/acc.cgi?acc=GSE94682 | NCBI Gene Expression Omnibus, GSE94682 |

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
