## [Decision Letter]

**Acceptance summary:**

Androgen receptor (AR) inhibitors are commonly used to treat prostate cancer, but resistance to these drugs is a problem. Up-regulation of the glucocorticoid receptor (GR) is associated with drug resistance. This paper provides a mechanistic link between the transcriptional co-repressor TLE3 and GR-mediated resistance to AR inhibitors in a prostate cancer cell line. This link involves TLE3 and AR binding to the GR locus as well as GR binding to TLE3/AR regulated genes. An association between TLE3 and GR expression in patient samples suggests the possibility of clinical relevance.

**Decision letter after peer review:**

Thank you for sending your article entitled "TLE3 loss confers AR inhibitor resistance by facilitating GR-mediated prostate cancer cell growth" for peer review at *eLife*. Your article is being evaluated by three peer reviewers, and the evaluation is being overseen by a guest Reviewing Editor and Kevin Struhl as the Senior Editor.

Summary:

While the reviewers found your work to be of generally high quality, there were concerns about the lack of adequate novel mechanistic insights. There are two main issues. First, there is a worry that the results are observed in a single cell line and that other cell lines behave differently. Thus, the reviewers are interested in additional mechanistic understanding, particularly with respect to this cell-line specificity. Second, given the single cell line issue, the reviewers request some connection of the mechanism shown here with human cancer. For example, is there any evidence for cancer relevance of the pathway in the various databases of human cancer?

Reviewer #1:

This is a very interesting study by Palit et al. that reports on the loss of the transcription factor co-repressor, TLE3, by CRISPR-library screening that renders LNCaP cells (not any of the other tested CRPC lines) resistant to AR-antagonist treatment with Enzalutamide or Apalutamide. TLE3 appears to be itself negatively regulated by AR. The initial findings were followed by RNA-seq assessment of TLE3 KO cells treated with Enz versus vehicle and led to the identification of GR as one of the top differentially expressed genes. The network analysis indicated that TLE3 KO facilitates reactivation of AR-targeted genes under Enz treatment, suggesting involvement of GR as the most likely mechanism of AR-bypass. Both pharmacological and genetic inhibition of GR was sufficient to reverse the resistant phenotype acquired by TLE3 KO. While the phenotypic assessments of TLE3's role in the context of AR and GR inhibition are promising with clear results, the study lacks sufficient mechanistic exploration and evidence for a conclusive story.

Major comments:

1) Please provide GR protein expression data wherever the transcript levels are analyzed because this is purported to be a critical part of the mechanistic link.

2) The CRISPR-screen was performed with treatment and selection of CRISPR-targeted Enz-naïve cells for 6 weeks. However, the emergence and expansion of resistant WT LNCaP cells takes considerably longer. Is TLE3 downregulated in parental cells that become Enz-resistant, when compared to Enz-naïve cells? In other words, is TLE3 depletion naturally required to allow for emergence of Enz resistance?

3) To conclude that depletion of TLE3 frees AR binding regions for GR occupancy (as in the Figure 5 model), the authors would require genome wide ChIP-seq of AR, GR and TLE3 in Enz-treated control vs *TLE3^KO^* cells. While the results clearly indicate an enhanced GR induction as a result of TLE3 depletion, no other data presented appears to support TLE3 inhibition as a pre-requisite for GR occupancy.

4) TLE3 is a known inhibitor of the Wnt pathway whose activation has been reported by several groups to be essential for Enz-resistance. Is it possible that TLE3 loss functions through the Wnt pathway (e.g., via β-catenin) instead of stimulating GR expression/action? Have the authors tested this alternative hypothesis?

5) The TCGA patient dataset is not well described. TCGA data are mainly obtained from local prostate cancer and biochemical recurrence usually refers to recurrence after local therapy (i.e., surgery or radiation), which is usually not relevant to hormone therapy resistance. What is the "anti-hormonal therapy" the authors refer to? How was the expression cut point defined and identified?

6) Overall, there appears to be insufficient data to support the model proposed in Figure 5.

Reviewer #2:

A couple previous studies have indicated an association between TLE3 and AR, but its functional significance for AR function has not been determined. This study found that TLE3 loss enhanced LNCaP cell growth in response to AR antagonist in a CRISPR screen. Further ChIP-seq and transcriptome data support TLE3 coregulation of a subset of AR regulated genes. Finally, in a TLE3 KO background the investigators found that AR inhibition caused an increase in GR, and that GR inhibition could resensitize the cells to ENZ. Together these data support a role for TLE3 in the regulation of a subset of AR regulated genes. However, there are a number of concerns related to functional significance that should be addressed, as indicated below.

1) The generality of the findings is unclear as TLE3 depletion only conferred drug resistance in LNCaP, but not in CWR-R1 or LAPC4. It would be of value to explore the basis for this difference a bit more. In particular, does ENZ stimulate TLE3 in these latter cells.

2) The significance would also certainly be enhanced by in vivo data confirming that the TLE3 depleted cells are resistant to ENZ.

3) AR was the top TF associated with genes that were differentially expressed in TLE3 KO cells in the presence of androgen or with ENZ. One would predict that expression of these genes would increase in the TLE3 KO cells. Figure 2B shows fold change, but it is not clear what is being compared. Moreover, the description is somewhat cryptic (genes that show an interaction with TLE3 KO and ENZ treatment). The effect of TLE3 on AR/TLE3 shared genes, as well as on TLE3 alone genes should be clearly described.

4) It is not clear how UGT2B17 fits with the hypothesis that TLE3 is repressing AR regulated genes in response to ENZ (Figure 2B).

5) It is an attractive hypothesis that AR directly represses TLE3 expression. However, supplemental Figure 2 only shows effects on protein, with no indication of how long the cells were treated. The authors should show a time course of TLE3 mRNA induction and loss in response to ENZ and DHT in order to address whether the effects are likely direct.

6) Figure 3 seems to show that TLE3 binding at AR regulated genes does not decrease with R1881 stimulation. This would seem to be inconsistent with the feedback model, and with the marked decrease in TLE3 protein in R1881 treated cells.

7) The growth data in Figure 4 is qualitative and only a single plate is shown for each condition. It should be quantified, and a growth curve would help.

8) The authors show that GR KO confers sensitivity to ENZ in the TLE3 KO cells, and suggest this reflects GR activation of AR/TLE3 regulated genes. To assess this mechanism, they should address whether GR KO in this context does indeed suppress the expression of AR/TLE3 coregulated genes.

9) While GR may contribute to ENZ resistance in TLE3 KO cells, the significance of this finding for TLE3 intact cells is not addressed. Is there increased GR recruitment to AR/TLE3 regulated genes in response to ENZ in wild-type cells?

10) In Supplemental Figure 4 it is unclear if the authors are assessing BCR after RP or after ADT. It is probably the former, which would only mean that TLE3 expression is associated with aggressiveness, and would not provide evidence that it is involved with response to ADT.

Reviewer #3:

The authors identify TLE3 loss as a cause of apalutamide resistance in a genome-wide in vitro CRISPR screen in LNCaP prostate cancer cells. Transcriptomic and ChIP-seq studies support a model whereby TLE3 loss rescues suppression of AR pathway signaling by antiandrogens. Mechanistically this occurs, at least in part, through upregulation of GR based on experiments showing that shRNAs targeting GR restore sensitivity to enzalutamide. This model is supported by ChIP-seq data showing overlapping binding of TLE3 and FOXA1 at various AREs across the genome including a GR enhancer.

The data supporting TLE3 as screen hit as well as the proposed mechanism for causing antiandrogen resistance through GR upregulation,is convincing, but the work has two significant shortcomings.

1) The apparent context dependence of TLE3 loss for LNCAP cells only (negative results in CWR-R1 and LAPC4) raises concerns about the broader relevance of the TLE3 loss in prostate cancer. The authors could address this in several ways:

i) testing of more models

ii) mechanistic insight into why resistance is not seen in CWR-R1 and LAPC4 (does TLE3 loss cause similar perturbations in AR signaling in these models?)

iii) deeper interrogation of TLE3 status in human datasets, particularly in the castration-resistant setting (several are now available from SU2C Prostate Dream team projects)

2) The prior work on GR in castration resistance diminishes the novelty. This could perhaps be overcome with additional mechanistic insight beyond that reported in the earlier publications. For example, how does TLE3 loss impact the chromatin landscape (particularly repressive histone marks) across the genome and more specifically at the GR locus?

Overall I would be supportive of considering a revised manuscript but it would need to have additional data.

[Editors' note: further revisions were requested prior to acceptance, as described below.]

Thank you for resubmitting your work entitled "TLE3 loss confers AR inhibitor resistance by facilitating GR-mediated human prostate cancer cell growth" for further consideration by *eLife*. Your revised article has been evaluated by Kevin Struhl (Senior Editor) and a guest Reviewing Editor.

The manuscript has been improved but there are some remaining issues that need to be addressed before acceptance, as outlined below:

Please revise the manuscript as requested by the reviewers.

Reviewer #1:

Palit, et al., provide a revised manuscript on TLE3 loss and enzalutamide resistance. The strength of the original manuscript is with the enzalutamide resistant phenotype conferred by TLE3 KO but was lacking in mechanistic characterization and there were some concerns about clinical relevance of this finding. In general, the authors have put forth a reasonable effort to address reviewer comments.

The additional evaluation on inverse correlation between GR and TLE3 expression from clinical data sets (Figure 5A and 5B) are not really convincing and requested in vivo experiments by one of the other reviewers on TLE3 KO and resistance are not provided. Together, with a CRISPR KO generated mechanism of resistance that occurs in a single cell line model, if moving forward with this manuscript I would suggest that the authors pull back on the potential clinical relevance and explicitly state this as a major caveat of their story (in the Abstract and Discussion).

Reviewer #2:

The authors have responded to the points raised in the initial review. There are still concerns that the observations are limited to one cell line and are only in vitro. However, the authors do now show an inverse correlation between TLE3 and GR in public domain data. It should be noted that the previous study from Arora et al. found that GR was not immediately increased by ENZ, but required some substantial time for adaptation. Therefore, overall the data support the hypothesis that loss of TLE3 is a mechanism that contributes to increasing GR in response to ENZ in at least a subset of patients. One minor point is that in response to the question of whether TLE3 loss increased WNT signaling, the authors showed no increase in active b-catenin. However, TLE3 loss would presumably increase TCF activity without an increase in active b-catenin. The more relevant data would be effects on TCF regulated genes such as *AXIN2*.

Reviewer #3:

The authors have responded to my earlier review with additional evidence from clinical datasets that extends the TLE3 work beyond the LNCaP cell line. I believe the revised manuscript is suitable for publication at *eLife*.

---

## [Author Response]

Reviewer #1:

This is a very interesting study by Palit et al. that reports on the loss of the transcription factor co-repressor, TLE3, by CRISPR-library screening that renders LNCaP cells (not any of the other tested CRPC lines) resistant to AR-antagonist treatment with Enzalutamide or Apalutamide. TLE3 appears to be itself negatively regulated by AR. The initial findings were followed by RNA-seq assessment of TLE3 KO cells treated with Enz versus vehicle and led to the identification of GR as one of the top differentially expressed genes. The network analysis indicated that TLE3 KO facilitates reactivation of AR-targeted genes under Enz treatment, suggesting involvement of GR as the most likely mechanism of AR-bypass. Both pharmacological and genetic inhibition of GR was sufficient to reverse the resistant phenotype acquired by TLE3 KO. While the phenotypic assessments of TLE3's role in the context of AR and GR inhibition are promising with clear results, the study lacks sufficient mechanistic exploration and evidence for a conclusive story.

We thank the reviewer for the constructive comments, and are delighted that the reviewer found our manuscript very interesting and promising with clear results. As suggested, we now provide more mechanistic explanation in the revised version, which we hope the reviewer appreciates.

Major comments:1) Please provide GR protein expression data wherever the transcript levels are analyzed because this is purported to be a critical part of the mechanistic link.

This is an excellent point, and we thank the reviewer for this constructive suggestion. As requested, we now added a western blot analysis (Figure 4A) showing protein expression levels of GR in control and *TLE3^KO^* cells treated with vehicle or enzalutamide for 5 days. This experiment shows clear upregulation of GR on protein level only in *TLE3^KO^* cells when treated with enzalutamide. These findings are in full concordance with our RNA-seq and qPCR-based conclusions that GR levels were increased upon Enzalutamide treatment when TLE3 was lost.

2) The CRISPR-screen was performed with treatment and selection of CRISPR-targeted Enz-naïve cells for 6 weeks. However, the emergence and expansion of resistant WT LNCaP cells takes considerably longer. Is TLE3 downregulated in parental cells that become Enz-resistant, when compared to Enz-naïve cells? In other words, is TLE3 depletion naturally required to allow for emergence of Enz resistance?

This is an interesting suggestion. We tested this hypothesis by assessing TLE3 protein levels in a panel of enzalutamide-resistant and parental cell lines (see Author response image 1). Interestingly, in none of the cell lines tested, TLE3 was found downregulated. These data suggest that TLE3 decrease is not a bona fide intrinsic feature of Enzalutamide resistance. That said, multiple different mechanisms have been reported to date, in which the TLE3-GR axis would represent one of these possible mechanisms. The number of cell line model systems available with acquired Enzalutamide resistance is limited, and we therefore cannot indisputably make a claim whether or not cell lines would be identified with this resistance mechanism. We would like to highlight that our newly added results in patient samples show that TLE3 and GR are inversely correlated in prostate tumors (Figure 5A,B) and upon enzalutamide resistance TLE3 is lost while GR is increased in some -but not all- paired tumor samples (Figure 5D).

**Author response image 1. respfig1:** TLE3 protein expression levels in indicated cell lines (untreated). Vinculin was used as a loading control.

3) To conclude that depletion of TLE3 frees AR binding regions for GR occupancy (as in the Figure 5 model), the authors would require genome wide ChIP-seq of AR, GR and TLE3 in Enz-treated control vs TLE3^KO^ cells. While the results clearly indicate an enhanced GR induction as a result of TLE3 depletion, no other data presented appears to support TLE3 inhibition as a pre-requisite for GR occupancy.

We thank the reviewer for bringing up this point and valuable suggestion. We fully agree that investigating GR occupancy would provide important evidence that would further support the model. To address this question, we performed ChIP-qPCR for GR at the loci of several target genes (RND3, GNAI1 and TNFRSF19) that were identified in the transcriptomics analyses in Figure 2. We found that GR only occupies the enhancer elements of these genes in the context of TLE3 loss combined with enzalutamide treatment (Figure 4G), which is in line with western blot results showing only GR expression in this context (Figure 4A). Combined, these experiments support our model that TLE3 inhibition in combination with AR inhibition is a pre-requisite for GR protein upregulation and enhancer occupancy.

4) TLE3 is a known inhibitor of the Wnt pathway whose activation has been reported by several groups to be essential for Enz-resistance. Is it possible that TLE3 loss functions through the Wnt pathway (e.g., via β-catenin) instead of stimulating GR expression/action? Have the authors tested this alternative hypothesis?

This is a very interesting hypothesis, and we thank the reviewer for bringing this to our attention. As suggested by the reviewer, we determined the levels of active ß-catenin upon TLE3 loss in combination with vehicle or enzalutamide treatment (Figure 2—figure supplement 1A), indicating that the resistance mediated by TLE3 loss is not the result of increased Wnt activation.

5) The TCGA patient dataset is not well described. TCGA data are mainly obtained from local prostate cancer and biochemical recurrence usually refers to recurrence after local therapy (i.e., surgery or radiation), which is usually not relevant to hormone therapy resistance. What is the "anti-hormonal therapy" the authors refer to? How was the expression cut point defined and identified?

We apologize for this omission, which is now resolved. Indeed, as the TCGA cohort represents tumors with localized disease, the vast majority of cases did not receive any adjuvant hormone therapy. However, a subset of 65 patients in the entire TCGA cohort did receive hormonal therapy, and the data presented in our manuscript is specifically describing these patients. Unfortunately, the clinical information in the TCGA cohort was insufficiently detailed that no specifics were provided on which type of hormonal therapy was prescribed. Regarding the definition of the expression cut-off in the TCGA analyses, we investigated the 3rd-4th-5th-6th-7th deciles, in which the 3rd decile provided the most-statistically significant difference on biochemical relapse-free survival. This information has now been included in the Materials and methods section.

To further strengthen the clinical part of our study, we now include additional analyses from two independent cohorts (new Figure 5A,B) showing an inverse correlation between TLE3 and GR mRNA levels. In addition, we now included new data generated from an on-site clinical study, in which biopsies were taken from metastatic lesions before and after enzalutamide exposure (new Figure 5D). These findings show that for a subset of patient samples we analyzed, TLE3 levels were found decreased upon relapse to Enzalutamide treatment, which was accompanied by an increase of GR in these samples.

6) Overall, there appears to be insufficient data to support the model proposed in Figure 5.

We thank the reviewer for highlighting this point. We have updated the manuscript with new data ensuring every step in the model is fully experimentally supported, either by us or by previous studies from others. In this revised manuscript, we included several analyses that support our model and have improved this study:

– We overlaid the ChIP-seq data derived by Stelloo et al. with data from Shah et al. of the GR enhancer unit and found TLE3 binds at the same locus.

– We show that GR is expressed on protein level in *TLE3^KO^* cells treated with enzalutamide. In these conditions, GR binds enhancer elements proximal to the TLE3/AR bound genes most differentially expressed in control cells versus *TLE3^KO^* cells treated with enzalutamide.

– We analyzed patient biopsy samples pre- and post-enzalutamide treatment for TLE3 and GR expression and observed low TLE3 and high GR expression in several cases of enzalutamide resistance. Furthermore, we analyzed publicly available RNA-seq data revealing an inverse correlation between TLE3 expression and GR expression in biopsy samples from prostate cancer patients with early-stage disease as well as advanced prostate cancer.

We hope that by addressing the reviewers comments, and including these additional analyses, the reviewer will find this manuscript suitable for publication *eLife*. In addition, we feel that our newly added experiments and analyses aided in filling the omissions in the model, and we sincerely hope the reviewer agrees.

Reviewer #2:

A couple previous studies have indicated an association between TLE3 and AR, but its functional significance for AR function has not been determined. This study found that TLE3 loss enhanced LNCaP cell growth in response to AR antagonist in a CRISPR screen. Further ChIP-seq and transcriptome data support TLE3 coregulation of a subset of AR regulated genes. Finally, in a TLE3 KO background the investigators found that AR inhibition caused an increase in GR, and that GR inhibition could resensitize the cells to ENZ. Together these data support a role for TLE3 in the regulation of a subset of AR regulated genes. However, there are a number of concerns related to functional significance that should be addressed, as indicated below.1) The generality of the findings is unclear as TLE3 depletion only conferred drug resistance in LNCaP, but not in CWR-R1 or LAPC4. It would be of value to explore the basis for this difference a bit more. In particular, does ENZ stimulate TLE3 in these latter cells.

The reviewer raises an excellent point. Indeed, we were also surprised to see that the TLE3-mediated enzalutamide mechanism is not conserved in CWR-R1 and LAPC4 cells. However, as we did find this to be crucial information, we decided to incorporate these results into the manuscript. As suggested by the reviewer, we now added Western blot results for TLE3 expression in LAPC4 and CWR-R1 cells treated with vehicle, R1881 or AR inhibitor (Figure 2—figure supplement 1F). The results show a difference in TLE3 expression levels in LAPC4 cells treated with enzalutamide and R1881 following the same direction as observed in LNCaP cells, though to a lesser extent. This effect was observed to be noticeable but weak in CWR-R1 cells. Potentially, this difference in androgen-regulated TLE3 expression levels could indeed explain the difference regarding the enzalutamide resistance observed in LNCaP and not LAPC4 and CWR-R1.

Although additional cell lines showing the same effect of TLE3 loss would be preferred, we do not expect this resistance mechanism to be generally applicable, as there is considerable heterogeneity in resistance mechanisms in prostate cancer patients.

Nevertheless, to demonstrate more general applicability of our findings, we tested TLE3 and GR expression by immunohistochemistry in paired biopsies of prostate cancer patients treated with enzalutamide. These analyses revealed low TLE3 expression and high GR expression in several cases of enzalutamide resistance, thus providing evidence for more general applicability of our findings.

2) The significance would also certainly be enhanced by in vivo data confirming that the TLE3 depleted cells are resistant to ENZ.

We fully agree with the reviewer, that in vivo data would greatly contribute to the significance of our work. To address this, we included two new types of analyses on human prostate tumor datasets. First, we included additional analyses on publicly-available transcriptomics data from two independent prostate cancer cohorts (new Figure 5A,B), in which an inverse correlation was observed between GR and TLE3 mRNA levels in vivo, strengthening our cell line-based findings that TLE3 negatively regulated GR transcription. Second, we incorporated data from an on-site clinical trial, in which biopsies from metastatic lesions were taken before enzalutamide treatment and after progression (new Figure 5D and Figure 5—figure supplement 1A). As expected, heterogeneity is found between patients, as multiple different mechanisms of enzalutamide resistance can occur. For example, one patient (out of 4) showed clear AR amplification arising during enzalutamide treatment. Nevertheless, for the other three patients, we observed low TLE3 and high GR expression associated with resistance. For one of these patients the inverse association between TLE3 and GR became more pronounced upon enzalutamide treatment (Figure 5D). These findings are in full concordance with our cell line-based experiments and illustrate that our findings can be recapitalized in vivo.

3) AR was the top TF associated with genes that were differentially expressed in TLE3 KO cells in the presence of androgen or with ENZ. One would predict that expression of these genes would increase in the TLE3 KO cells. Figure 2B shows fold change, but it is not clear what is being compared. Moreover, the description is somewhat cryptic (genes that show an interaction with TLE3 KO and ENZ treatment). The effect of TLE3 on AR/TLE3 shared genes, as well as on TLE3 alone genes should be clearly described.

We thank the reviewer for highlighting this unclarity in the manuscript, which was also raised by reviewer #1. To address this point, we now reformulated the text and figure legend to describe which groups are being compared (control cells versus *TLE3^KO^* cells treated with enzalutamide). Through this comparison, Figure 2B shows that loss of TLE3 leads to reactivation of AR-driven genes in the presence of enzalutamide, which does not happen in TLE3-proficient cells. We changed the term “interaction” into “correlation” in the manuscript.

4) It is not clear how UGT2B17 fits with the hypothesis that TLE3 is repressing AR regulated genes in response to ENZ (Figure 2B).

We apologize for the lack of clarity. UGT2B17 is negatively regulated by AR which is illustrated by the fact that expression of this gene goes up upon enzalutamide treatment rather than down. Loss of TLE3 appears to dampen this upregulation, similar to dampening downregulation of genes positively regulated by AR when cells are treated with enzalutamide. In the Results section, this is now more-clearly explained and explicitly mentioned.

5) It is an attractive hypothesis that AR directly represses TLE3 expression. However, supplemental Figure 2 only shows effects on protein, with no indication of how long the cells were treated. The authors should show a time course of TLE3 mRNA induction and loss in response to ENZ and DHT in order to address whether the effects are likely direct.

The reviewer raises an excellent point. To address this, we analyzed publicly available RNA-seq data (Massie et al., 2011) from LNCaP cells treated with R1881 (time course) for TLE3 expression. These data have been added to Figure 2—figure supplement 1H. The results show that TLE3 mRNA levels decrease rapidly, as soon as 4 hours, after stimulation with R1881, providing evidence that AR directly represses TLE3 transcription. These data are in line with the western blot results (Figure 2—figure supplement 1E) and provide evidence that AR directly represses TLE3 transcription.

6) Figure 3 seems to show that TLE3 binding at AR regulated genes does not decrease with R1881 stimulation. This would seem to be inconsistent with the feedback model, and with the marked decrease in TLE3 protein in R1881 treated cells.

Indeed, as TLE3 is suppressed by AR activation, persistence of TLE3 binding at AR regulated genes after R1881 could potentially be in contrast to the model. However, we would like to highlight the aspect of protein stability and turnover as a variable. In Figure 3, the ChIP-seq was performed using cells that were exposed to R1881 for 4 hours to facilitate AR translocation to the nucleus and chromatin binding(Jariwala et al., 2007; Tewari et al., 2012). At this timepoint, TLE3 protein levels are not yet affected (as was shown in Stelloo et al., 2018 Oncogene, Figure 1C).Therefore, we believe these findings do not contradict the model. We updated the Discussion section, and explicitly refer to the data from Stelloo et al. to further explain this aspect.

7) The growth data in Figure 4 is qualitative and only a single plate is shown for each condition. It should be quantified, and a growth curve would help.

We agree that quantitative data for the growth assays would be superior over the mere qualitative data as provided in the original paper. To address this point, we now quantified the data for the growth assays shown for this figure (Figure 4—figure supplement 1C). The conclusion remains identical to the original observations, and statistical testing confirmed that there is significant difference in growth between control and *TLE3^KO^* cells treated with enzalutamide.

8) The authors show that GR KO confers sensitivity to ENZ in the TLE3 KO cells, and suggest this reflects GR activation of AR/TLE3 regulated genes. To assess this mechanism, they should address whether GR KO in this context does indeed suppress the expression of AR/TLE3 coregulated genes.

In the original manuscript, we performed shRNA targeting of TLE3. Using the same model system for RT-QPCR analyses on AR/TLE3 regulated genes, we were unable to consistently detect impact of GR knockdown on expression of these genes, suggesting that the remaining low levels of GR sufficed to preserve the phenotype on a transcriptomics level. In order to address the reviewer question, we reverted to the original observation from the Sawyers lab (Arora et al., 2013), in which these very same genes as we study in our paper (RND3, GNAI1, GR, UGT2B17 and PMP22) (Figure 4—figure supplement 1E) were tested for expression after targeting GR. Thus, for several of the most differentially expressed TLE3/AR target genes we identified (Figure 2B,) GR was shown to be functionally and critically involved in the regulation of these genes, as was originally shown by the Charles Sawyers lab. We highlighted this in the Results and Discussion sections.

9) While GR may contribute to ENZ resistance in TLE3 KO cells, the significance of this finding for TLE3 intact cells is not addressed. Is there increased GR recruitment to AR/TLE3 regulated genes in response to ENZ in wild-type cells?

While we agree it would be interesting to investigate GR occupancy in this context, GR appears to be only upregulated when both TLE3 and AR are perturbed in our model as shown in Figure 2. Newly added western blot results (Figure 4A) show no detectable GR protein in WT cells treated with enzalutamide, making GR chromatin occupancy unlikely in this context. We updated the Discussion section to highlight this point. Our newly added GR ChIP-qPCR data (Figure 4G) confirms this, as GR occupancy is only observed proximal to AR/TLE3 genes in *TLE3^KO^* cells treated with enzalutamide, but not in untreated- or enzalutamide-treated WT cells.

10) In Supplemental Figure 4 it is unclear if the authors are assessing BCR after RP or after ADT. It is probably the former, which would only mean that TLE3 expression is associated with aggressiveness, and would not provide evidence that it is involved with response to ADT.

We apologize for the unclarity. Indeed, as the TCGA cohort represents tumor with localized disease, the vast majority of cases did not receive any adjuvant hormone therapy. However, a subset of 65 patients in the entire TCGA cohort did receive hormonal therapy, and the data presented in our manuscript is specifically describing these patients. As such, the BCR as described in Figure 5C is after hormonal therapy. Unfortunately, the clinical information in the TCGA cohort was insufficiently detailed in the sense that no specifics were provided on which type of hormonal therapy was prescribed.

We further strengthened the connection between TLE3 and Enzalutamide resistance in two ways, as we mention in reply to point 2 of this reviewer: 1) correction of TLE3 and GR mRNA levels in 2 prostate cancer cohorts. 2) Decrease of TLE3 levels and increased GR levels in patient samples relapsing after enzalutamide in an on-site performed clinical trial. We believe these new additions further strengthen the clinical implications of our work, and we sincerely hope the reviewer agrees.

Reviewer #3:

The authors identify TLE3 loss as a cause of apalutamide resistance in a genome-wide in vitro CRISPR screen in LNCaP prostate cancer cells. Transcriptomic and ChIP-seq studies support a model whereby TLE3 loss rescues suppression of AR pathway signaling by antiandrogens. Mechanistically this occurs, at least in part, through upregulation of GR based on experiments showing that shRNAs targeting GR restore sensitivity to enzalutamide. This model is supported by ChIP-seq data showing overlapping binding of TLE3 and FOXA1 at various AREs across the genome including a GR enhancer.The data supporting TLE3 as screen hit as well as the proposed mechanism for causing antiandrogen resistance through GR upregulation,is convincing, but the work has two significant shortcomings.

We thank the reviewer for the constructive comments and valuable suggestions. As further elaborated below, we did our utmost best to address the two issues what were raised by this reviewer.

1) The apparent context dependence of TLE3 loss for LNCAP cells only (negative results in CWR-R1 and LAPC4) raises concerns about the broader relevance of the TLE3 loss in prostate cancer. The authors could address this in several ways:i) testing of more modelsii) mechanistic insight into why resistance is not seen in CWR-R1 and LAPC4 (does TLE3 loss cause similar perturbations in AR signaling in these models?)iii) deeper interrogation of TLE3 status in human datasets, particularly in the castration-resistant setting (several are now available from SU2C Prostate Dream team projects)

We thank the reviewer for highlighting this critical point. Indeed, the impact of TLE3 loss in prostate cancer cells was quite context dependent, and the observations in LNCaP cells could not be recapitulated in CWR-R1 and LAPC4 cells. However, as we did find this to be crucial information, we decided to incorporate these results into the manuscript. To provide more insights into this discrepancy, we now added Western blot results for TLE3 expression in LAPC4 and CWR-R1 cells treated with vehicle, R1881 or AR inhibitor (Figure 2—figure supplement 1F). The results show a difference in TLE3 expression levels in LAPC4 cells treated with enzalutamide and R1881 following the same direction as observed in LNCaP cells, though to a lesser extent. This effect is far less prominent in CWR-R1 cells. Potentially, this difference in androgen-regulated TLE3 expression levels could indeed explain the difference regarding the enzalutamide resistance observed in LNCaP and not LAPC4 and CWR-R1

To address the reviewer’s concern regarding human datasets, we significantly increased the amount of data describing TLE3 in human datasets. We now include additional analyses from two independent prostate cancer patient cohorts (new Figure 5A,B), showing an inverse correlation between TLE3 and GR mRNA levels. In addition, we generated important additional data from an on-site clinical trial, in which biopsies were taken from metastatic lesions before and after enzalutamide exposure (new Figure 5D and Figure 5—figure supplement 1A). These analyses revealed low TLE3 expression and high GR expression in several cases of enzalutamide resistance.

With these new data, we sincerely hope the reviewer finds this issue sufficiently addressed.

2) The prior work on GR in castration resistance diminishes the novelty. This could perhaps be overcome with additional mechanistic insight beyond that reported in the earlier publications. For example, how does TLE3 loss impact the chromatin landscape (particularly repressive histone marks) across the genome and more specifically at the GR locus?

Indeed, prior work from the Sawyers group showed that GR can mediate Enzalutamide resistance. However, this previous work did not explain how GR upregulation was mediated. Our results show that TLE3 acts as a transcriptional repressor that plays a critical role in inhibiting the expression of GR in prostate cancer cells. Upon TLE3 loss and AR inhibition, this suppression is alleviated, resulting in an increase of GR levels, both on the transcript (Figure 2B,C) and protein levels (Figure 4A), giving rise to Enzalutamide resistance. With this, we feel that our findings do provide substantial novelty, in explaining how GR upregulation in Enzalutamide resistance can occur: through a loss of the transcriptional repressive effects of TLE3. We have now updated the Discussion section to communicate this message more clearly.

To address the question how loss of TLE3 affects the chromatin at the GR locus, we performed H3K27ac ChIP-qPCR for the GR enhancer occupied by TLE3. TLE3 is known to facilitate a repressed chromatin structure by recruiting HDACs. Loss of TLE3 was shown to positively regulate H3K27 acetylation in breast cancer cells, affecting chromatin structure at TLE3/ERα binding sites resulting in derepression of these loci/genes (Jangal et al., 2014).

In our model, loss of TLE3 resulted in upregulation of H3K27ac when compared to WT cells (Figure 4—figure supplement 1D). We also looked beyond the GR enhancer by analyzing the RND3 locus as this gene showed upregulation upon TLE3 loss. Also here we found that H3K27 acetylation was increased significantly in *TLE3^KO^* cells when compared to TLE3-intact cells (Figure 4—figure supplement 1D). These data are in line with the expression data for RND3 in Figure 2, and newly added ChIP-qPCR data showing binding of GR proximal to RND3 and several other target genes despite enzalutamide treatment (Figure 4G).

Overall I would be supportive of considering a revised manuscript but it would need to have additional data.

We thank the reviewer for the constructive remarks and hope that these new findings on the transcriptional regulation of the GR locus, along with the new clinical datasets that strengthen the GR/TLE3 connection, sufficiently address the points raised by this reviewer and would render our manuscript acceptable for publication.

To summarize, our most important new data includes:

– Addition of TLE3 and GR expression analysis in prostate cancer patient samples.

– Analysis of publicly available RNA-seq data analyzing TLE3 and GR expression in prostate cancer cohorts

– GR protein expression by western blot in control and *TLE3^KO^* cells treated with enzalutamide and TLE3 expression in LAPC4 and CWR-R1 cells in various conditions

– Analysis of time course of TLE3 expression in publicly available RNA-seq data (Massie et al. 2011) from LNCaP cells treated with R1881

– ChIP-qPCR analysis for the TLE3-mediated H3K27 acetylation at the GR locus in WT and *TLE3^KO^* cells, as well as ChIP-qPCR-analysis for GR at several other TLE3/AR target genes.

[Editors' note: further revisions were requested prior to acceptance, as described below.]

Reviewer #1:

Palit, et al., provide a revised manuscript on TLE3 loss and enzalutamide resistance. The strength of the original manuscript is with the enzalutamide resistant phenotype conferred by TLE3 KO but was lacking in mechanistic characterization and there were some concerns about clinical relevance of this finding. In general, the authors have put forth a reasonable effort to address reviewer comments.The additional evaluation on inverse correlation between GR and TLE3 expression from clinical data sets (Figure 5A and 5B) are not really convincing and requested in vivo experiments by one of the other reviewers on TLE3 KO and resistance are not provided. Together, with a CRISPR KO generated mechanism of resistance that occurs in a single cell line model, if moving forward with this manuscript I would suggest that the authors pull back on the potential clinical relevance and explicitly state this as a major caveat of their story (in the Abstract and Discussion).

We are delighted the reviewer appreciates the changes made to the original manuscript and we are thankful for the constructive comments helping us to further improve the manuscript. We have made changes to the Abstract and Discussion as requested. In the Abstract, we now mention that TLE3 and GR expression in clinical samples reflect our findings in LNCaP cells, but that clinical relevance is yet to be determined. This is also stated in the Discussion, where we (additionally) mention the limitations of the in vitro data. We state that our findings warrant further investigation into the clinical relevance of this resistance mechanism in patients.

We hope the reviewer appreciates the changes made to the revised manuscript and considers the manuscript suitable for publication at *eLife*.

Reviewer #2:

The authors have responded to the points raised in the initial review. There are still concerns that the observations are limited to one cell line and are only in vitro. However, the authors do now show an inverse correlation between TLE3 and GR in public domain data. It should be noted that the previous study from Arora et al. found that GR was not immediately increased by ENZ, but required some substantial time for adaptation. Therefore, overall the data support the hypothesis that loss of TLE3 is a mechanism that contributes to increasing GR in response to ENZ in at least a subset of patients. One minor point is that in response to the question of whether TLE3 loss increased WNT signaling, the authors showed no increase in active b-catenin. However, TLE3 loss would presumably increase TCF activity without an increase in active b-catenin. The more relevant data would be effects on TCF regulated genes such as AXIN2.

We are pleased that the reviewer appreciates the changes made to the manuscript in response to the comments made in the initial review. We thank reviewer#2 for the constructive comments and we now mention the adaptation time of the LREX model used in the study by Arora et al., 2013 in the Discussion.

To address the point regarding Wnt signaling through TCF activity mentioned by the reviewer, we now added qPCR data for the TCF target gene *AXIN2* in *TLE3^KO^* cells treated with vehicle or enzalutamide for 5 days (Figure 2—figure supplement 1B), and found no significant changes relative to control cells, indicating Wnt signaling is not altered in this context. We hope that our manuscript is now suitable for publication at *eLife*.

Reviewer #3:

The authors have responded to my earlier review with additional evidence from clinical datasets that extends the TLE3 work beyond the LNCaP cell line. I believe the revised manuscript is suitable for publication at eLife.

We thank the reviewer for the valuable suggestions and constructive comments and are pleased the reviewer appreciates the changes made to the article and believes that it is suitable for publication at *eLife*.